# What Is Happening in the Squares of China? Exploring the Experience of Participating in Square Sports and Sustainability Factors

**DOI:** 10.3390/ijerph192315693

**Published:** 2022-11-25

**Authors:** Guoyi Li, Jungrae Lee

**Affiliations:** 1Department of Physical Education, Shenyang Medical College, No. 146, Huanghe North Street, Shenyang 110034, China; 2Department of Leisure and Sports, College of Ecology and Environmental Science, Kyungpook National University, 2559, Gyeongsang-daero, Sangju-si 37224, Gyeongsangbuk-do, Republic of Korea

**Keywords:** square sports, China square, social mechanism influencing sustainability

## Abstract

This phenomenological study explored the experience of participation in square sports in China and the social mechanisms by which they can be sustained. Ten study participants were selected through a purposeful sampling method. The findings indicate that their physical and mental health were either maintained or improved as they engaged in square sports. They also experienced reduced feelings of loneliness and an increase in their sense of belonging through exchanges with other members of their teams. They enjoyed the freedom from cost and spatial restrictions in pursuing leisure activities. However, conflicts also arose with other groups, mainly related to securing space in the squares. Additionally, the study found that conflicts between participants and non-participants in square sports emerged as a social problem. The social mechanisms by which square sports can be sustained were identified as people-led voluntary participation, pride in square sports, and the reproduction of economic capital using human resources.

## 1. Introduction

A square refers to an open space in the center of a city where people gather in large numbers and move freely. It is a crucial city component, allowing citizens to enjoy daily leisure and sports activities simultaneously. In contemporary society, a square testifies a state’s history and the spirit of the times and signifies more than an empty space. This is because it does not have a fixed purpose and transforms itself according to the social atmosphere specific to an era. As a square also functions as a place where conflicting ideologies and class divisions are embraced and harmonized, which encourages people to lead an active leisure life, it reveals an era’s cultural phenomena and inherent characteristics [1].

In China, squares give rise to a unique form of culture. In the past, much like in other countries, squares were used only as venues for political actions. The examples include the anti-Japanese May Fourth Movement of 1919, the students’ protest movement of May 30, 1925, and Mao Zedong’s declaration of the founding of the People’s Republic of China in 1949. However, in recent years, the squares in China have been transformed into important areas that promote the citizens’ enjoyment, leisure, and mental and physical health.

Large groups of people dancing or exercising in the evening in the squares is a common sight in China. The physical activities, performed collectively in a square and ranging from the martial art form Tai Chi to disco and free-style dancing, are called square sports (It refers to all physical activities collectively performed along with music by a large number of people gathered in a square, such as gymnastics, walking, jumping rope, dancing, jianzi, and Tai Chi. There are various terms for these actions, such as square dance, square gymnastics, square dancing, etc. In this study, square sports refer to all physical activities performed in the squares). They can be considered a unique form of the popular leisure culture of the Chinese, which anyone can easily enjoy, regardless of age or sex, and is particularly popular among middle-aged women [2,3].

Why do Chinese people gather in squares to engage in collective physical activity? Square sports are closely linked with China’s economic and cultural policies. Since the People’s Republic of China was established in 1949, when the country changed from a Confucian to a socialist state, the government has obliged the people to gather in the squares and collectively engage in physical activities. The purpose of this initiative was to unite the students, educate them on socialism, and stimulate popular psychology by utilizing the ethos of a proletarian class culture [4].

As square sports emerged as a unique part of Chinese culture, they began to draw the attention of academia. Previous studies on this topic have focused on the evolution of square sports over the years [4,5,6,7,8], the effects of square sports on how a city is perceived in China [9,10,11,12], and the improvement in the participants’ quality of life as well as participation satisfaction [1,13,14,15,16,17,18,19]. However, most of the studies utilized a quantitative approach and very few have explored the lived experience of participants in square sports.

As mentioned earlier, the Chinese government made participation in square sports compulsory in 1980. The enthusiasm for square sports continues continuously throughout China [5]. A Guinness World Record was set in November 2016 when more than 50,000 people wearing identical clothes participated in the same physical activity in squares across 14 Chinese cities [6]. People are not forced to participate in square sports but voluntarily get involved. Even square dance activities have spread out and received widespread public attention over the world, such as France, New Zealand, and the United States. This phenomenon shows that square sports as a Chinese culture have been sustained. Culture has influential factors for sustainable development impacting social, economic, and environmental dimensions [19]. It is important to understand how China square sports have been sustained for long periods and what mechanism has been operated.

The phenomenological approach does not directly deal with the principles of social phenomena but seeks to understand the essence of human experience in society [20]. This study utilizes the phenomenological approach to explore why people continuously gather in squares, how they engage in collective physical activities, and what social mechanisms sustain the square sport in China’s culture.

## 2. Methods

This study used qualitative phenomenological methods, which are the most suitable for achieving the purpose of this study. The phenomenological approach explores individuals’ experiential meaning and understands the lived world. The main point of the phenomenological method is to understand the commonalities of individuals’ experiences so that this focuses on what and how an individual experiences a specific phenomenon [21,22,23]. Therefore, the phenomenological method is a critical approach to elucidate the meaning of lived experiences of a certain phenomenon for individuals with experiences. Given that this study aims to understand why people gather in squares, how they involve in square sports, and what square sports have been sustained, a phenomenological method is an appropriate approach to exploring a common phenomenon of what takes place in sports squares in China. In this regard, data were collected from study participants participating in square sports and detailed descriptions of the experiences shared with them were produced, regarding “what” they experienced and “how” they experienced it [20,21,24].

### 2.1. Research Environment

There are approximately 6000 squares in China, which are the settings for various daily physical activities. Shenyang is the largest city in the three northeastern provinces, and City Hall Square and Zhongshan Square, located in its center, are emblematic places. City Hall is the largest square in Shenyang with a surface area of 66,200 m^2^, whereas Zhongshan Square covers an area of 13,266 m^2^. They can accommodate a large number of people for sports activities daily.

### 2.2. Study Participants

The purposeful and snowball sampling method were used to select study participants who could provide necessary information about a certain phenomenon [21]. The sample size in the phenomenal research does not necessarily require large numbers of participants to comprehend individuals’ lived experiences. The typical sample sizes of the phenomenon study range from 1 to 10 persons [22]. The authors also did not observe new information and themes after interviewing 10 study participants. In this regard, the authors recruited 10 study participants, five of whom had been members of a certain team for at least two years and five were leaders of sports activity groups. The leaders were expected to provide insights into the experiences that the general participants did not share. Table 1 presents the personal characteristics of the study participants.

### 2.3. Data Collection

After approval (KNU-2022-0003) from Kyungpook National Institutional Review Board, the first author contacted the leader of a square sports team. It was not difficult to contact the leader since the first author was acquainted with them. He demonstrated the purpose of the study to the leader and had an opportunity to post a notice of recruitment for study participants in the chatroom of a social network service (SNS). He also requested those who were involved in the SNS to deliver the recruitment notice to others. Through this procedure, 10 study participants who met inclusion criteria (at least two years of experience) were recruited and asked to participate in this study.

This study conducted preliminary interviews with two study participants. This process helped the authors to determine the scope and content of the in-depth interviews. The authors interviewed 10 study participants using a mixture of unstructured and semi-structured interview questions.

Furthermore, this study conducted observations at City Hall Square and Zhongshan Square in China. In the observations, the authors were not active participants but look around people involved in square sports. The observation focused on how people gathered in squares, what sports activities existed in squares, how people occupied areas for square team sports, what conflict emerged, and how they coped with the conflict situation. The observations were conducted over 4 weeks with a total time of approximately 17 h. Field notes were written and used as supplementary data in this study.

In cases where further clarifications were necessary, additional interviews were conducted over the phone or by visiting the square. Journals, news articles, and specialized books related to Chinese square sports were used as supplementary data.

### 2.4. Data Analysis Method

After systematically transcribing the data obtained through in-depth interviews with the study participants, a textual analysis was performed. The next step was the open coding process through which the data were classified based on the subject by marking the ones that could offer important answers to the research questions while repeatedly reading the data, naming them, or writing down questions or thoughts that came to mind [22,23]. After that, the data with similar coded names were classified into higher categories. For example, words and phrases such as “conversations with other people and sharing intimacy” were classified as “reduction in loneliness and increase in the sense of belonging”. “Maintaining one’s place” and “conflicts over places” were classified as “conflicts with other teams to secure a space”. The data analysis was completed through a category-checking process to determine whether the finally constructed categories expressed all data related to the research questions [23].

### 2.5. Data Validation

Three strategies were applied to increase the validity of the analysis of the collected data and to secure truthfulness [24]. First, peer debriefing was conducted with three university professors specializing in sports sociology and policy. They checked all processes of data collection, analysis, and paper writing. Second, member checking was employed to accurately check the interpretation and themes that emerged through interviews. Third, triangulation was used to secure validity based on interviews, field notes, and journals [25,26].

## 3. Results and Discussion

This section discusses the experience of participation in square sports in China and its underlying social mechanisms. Figure 1 briefly shows the results of the study.

### 3.1. Experience of Participation in Square Sports

#### 3.1.1. Maintenance and Improvement of Physical and Mental Health

Various physical activities are undertaken in Chinese squares, such as health walking, aerobics, Tai Chi, dance sports, folk dance, jump rope, and jianzi. Square dancing, a popular form of square sports, has the longest history and the highest participation rate. It is a key event for understanding the Chinese square culture. An estimated 300 million people participate in square dancing throughout the country. This is expected to increase rapidly and exceed 500 million within a few years [1]. Square dancing is a type of gymnastic dance that has gradually gained popularity since the late 1980s. Those conscious about caring for their health consider it a cultural activity necessary to relieve stress, promote mental and physical stability, and build a harmonious society. Several studies have been conducted on the physiological effects of square dancing [27,28].


*My family worried that I might not be able to physically adapt to it because I had not exercised for a long time, but they were wrong. I used to be overweight and had high blood pressure. After several years of continuous exercise, I lost weight and my blood pressure normalized.*



*Square dancing is an exercise involving the whole body, which improves blood circulation and joint function in older individuals. It also reduces fat when performed steadily. In general, it is not strenuous and has appropriate intensity. Usually, low-intensity exercise for at least one hour improves the constitution of older individuals.*



*As people age, their reaction speed slows down, their concentration decreases, and their memory declines. They undergo psychological changes, such as developing eccentric personalities and emotional instability. However, participating in square sports for a long time made me feel that my emotions have become stable and optimistic. Middle-aged and older people can effectively overcome loneliness by participating in this practice.*


The study participants stated that their health improved through participation in square sports was the most important benefit. In particular, they mentioned weight loss and improvement in constitution and blood circulation as the most remarkable changes along with better mental health [29,30]. These results are consistent with the argument that square dancing is one of the most effective methods for improving constitution by promoting blood circulation in the body and improving tissue cells [31,32]. Similarly, another study reported that square dancing improved subjects’ constitution, reduced disease risk, and improved their sleep and body shape, leading to psychological stability [30].

China is one of the countries with a super-aging society. According to the National Bureau of Statistics, the average rate of increase in the aging population in China is 3.2% per year, and the older population growth rate is 5.21%, which is higher than the growth rate of the world population [33]. Thus, China faces various problems due to its increasing older population. Leisure activities have long been highlighted as a crucial social measure against psychological issues associated with physical illness, children’s marriage, reduction in living spaces and conversations, fear of death, and anxiety about the quality of life. In this respect, older adults’ participation in square sports is a desirable leisure activity.

#### 3.1.2. Reduction in Loneliness and Increase in the Sense of Belonging

According to the Chinese Family Development Report, older adults living alone account for approximately 10% of the total aging population, while older families account for approximately 41.9%. These figures are rapidly increasing [34] and represent the lives of older adults who have to live in isolation and for whom communication with society is an important purpose of living.


*As society develops, the sense of closeness between people gradually decreases. They no longer know who their neighbors are. When I was young, I had good relationships with neighbors, and I liked the feeling of living in proximity and helping each other. I found this feeling again while I was participating in square dancing. In addition to exercising, We go shopping, travel together, and stay friends, talking about the affairs around us—affairs of the state and the community. In addition, if someone has a problem, we help each other and even donate.*



*As urbanization expanded rapidly after the reform and opening up, people have come to form a cold society in cities, where they only look at the faces of their neighbors and do not even say hello. Therefore, we came to the square to exercise and find pleasure. We talk to each other while exercising and relieving our loneliness. We have also become closer to other people by eating together, traveling, and helping each other.*



*As I liked dancing originally, I formed a team for a long time, which motivated me to continue the dance I loved. This is nice. In my spare time, I could continue my favorite dance sport with my friends, and my body became healthy so that I did not have to burden my children. I would like to meet more friends and enjoy their company while working together.*


Most of those who participate in square sports are middle-aged or older. They had felt disconnected from their neighbors because of the progress of urbanization, which amplified loneliness. This phenomenon is one of the problems faced by middle-aged and older persons in most countries worldwide.

The study participants bonded with their neighbors through participation in square sports and thus experienced a sense of belonging. They engaged in sports activities and shared their private lives while shopping, traveling, and eating together. They also helped other participants in need, maintaining a sense of closeness and belonging.

It was found that square sports activities close the distance among people by creating environments where they can connect and enhance their ability to exchange and socialize with each other. As such, they enable them to control stress by expressing and relieving negative emotions, including tension, loneliness, and the sense of loss or emptiness. Eventually, participating in square sports activities significantly improves the quality of life of people [35,36]. These results support previous studies that participation in square team sports enhances happiness and promotes a healthy lifestyle [36] by restoring a sense of belonging and identity [37].

These activities are inclusive, and anyone can enjoy them. People can overcome the inconvenience of a constantly changing city by having fun social exchanges, building new relationships [38], and integrating into a new group.

#### 3.1.3. Freedom from Cost and Spatial Restrictions

A square is an open space for residents. It has no owner and can therefore be used by anyone. The participants stated that one of the critical aspects of participating in square sports is not being constrained by cost or space.


*Our village has gymnasiums and cultural palaces (e.g., fitness centers) managed by a state institution. However, it is difficult to make a reservation, and it is inconvenient to use them because there is a fee and time limit. However, squares can be used indefinitely and free of charge; therefore, they are more convenient.*



*Indoor exercise places or gymnasiums in rural areas are managed by government agencies and cannot be easily reserved. Few gymnasiums in schools can be opened to the public. Even when they could be reserved, they were expensive to use. Therefore, citizens began to choose squares. The cost was not very burdensome as it was only 30 CNY(Chinese Yuan Renminbi)per month, that is, 360 CNY per year.*



*As cities develop, the number of buildings and green areas where people can exercise gradually decreases. Therefore, people have begun to find other places to exercise to improve their physical health. These are mainly squares, parks, spaces in front of apartments, and sports centers. Squares have advantages such as large areas, good air quality, and unlimited use.*


Generally, participation in leisure activities requires an appropriate level of expenditure. However, such monthly expenses can be burdensome for those who do not engage in economic activities. In this respect, unlike other leisure activities, square sports are accessible to everyone regardless of their financial status. As such, the square is an appropriate place to spend a lot of time in the context of the contraction of economic activities and forced leisure after retirement. However, the most basic aspect of square sports is securing free space, which suggests that it may be difficult to satisfy the desires of all who wish to participate if there is no adequate space [39].

Another study found that [14] older people who engaged in group physical training in Zizhuyuan Park, Beijing, also felt frustrated about the fact that activity centers for older adults were spatially “blocked”. The study participants said they went to the park to enjoy the “fresh air”. The “fresh air” they were talking about was closer in meaning to the subjective sense of “an open space that is not stuffy” rather than avoiding pollution. In addition, they insisted on parks because they perceived activity centers for older people as expensive. Such places are open to people over the age of 60 because they are essentially built for the welfare of older adults; however, beneficiaries should pay usage fees ranging from five to ten yuan per month. Although cheaper than private sports facilities, most of those participating in square sports have never visited private facilities. They also believe that there is no need to do so as they can enjoy more diverse activities for free in the squares.

Consequently, those participating in square sports activities perceive squares as their own space that guarantees freedom from money and time. As such, square sports have become leisure activities that adequately satisfy the economic conditions and social relationships of older adults in China. In particular, the low cost and free use of spaces are great advantages for activating Chinese square sports.

#### 3.1.4. Conflicts with Other Teams to Secure a Space

Sharing space does not mean building space with a particular function or using it together. It should enable the formation of intimate relationships and interactions and lead to mutual exchanges [40]. However, sharing space for square sports is difficult since people focus on occupying a good place. It leads to various conflicts among participants in square sports. Various conflicts arise among participants in square sports, of which those about occupying a good place are the biggest problem.


*Before places are assigned, many conflicts occur, mainly because of the spaces for exercising. Each team thinks of various ways to maintain its place. Every time I come early, I occupy a place, draw boundaries, and exercise while wary of other team members and people around me. However, conflicts often arise. Our team was sometimes ignored because the number of team members was initially small.*



*Many people are in the square, and the number of teams is gradually increasing. Those who exercise alone do not exercise at fixed places but at every square corner. As cases occur where those who exercise alone move around or go back and forth between people who are exercising complaints are raised, leading to swearing. Conflicts often occur between nearby people due to the competition for places. People swear at each other daily because there is not enough space or quarrels due to disturbing and loud music.*



*People in the square were confused before their places were allocated. Those who exercised alone came to the square early to occupy more space, and teams that came late had many complaints because they had to exercise at the corners. They began coming earlier to occupy places the following day. Therefore, they had many complaints, swore first, and even fought later.*


The above statements demonstrate that the team leaders go to the square earlier than the general participants to install speakers and organize the environment depending on the planned activity. They had more conflict experiences due to space occupations than the other participants. This means that occupying a specific space in the square is directly related to the power of a team. In general, the power and force of a team help to occupy a certain place in a common space [41]. In this respect, occupying the center of the square has significance beyond the simple meaning of securing a place and becomes an indicator of rank and power among the square sports teams. Class inequality in modern society is shaped and reproduced through creation of space and its change [42]. Consequently, conflicts eventually affect the identity of the teams.

Team representatives discussed and decided to allocate locations using the lottery method to solve this problem. First, places were assigned numbers on pieces of paper drawn in the lottery. Teams that arrived later were added when spaces were available; otherwise, the teams were merged with others.

As Figure 2 shows, places were allocated through an agreement between team leaders. However, teams who were not given good places constantly complained, which led to conflicts. Because the square belongs to no one, the continuous struggle to occupy space can lead to inequality and irrationality. Although voluntarily organized square sports are in the limelight as part of the Chinese people’s cultural life, the unsystematic operation breeds disorder and conflicts in the square [43]. Therefore, an effective management system needs to be developed.

#### 3.1.5. Conflicts between Participants and Non-Participants in Square Sports

Square sports are leisure activities accessible to all Chinese citizens even under a poorly planned economy and are a heritage of the collective gymnastics culture from the days of socialism. Currently, approximately 100 million individuals voluntarily participate in sports. Public opinion on these activities is generally positive considering the benefits to the physical and mental health of elderly adults [44]. However, as with all social phenomena with pros and cons, conflicts with non-participants have also emerged as more people participate in square sports.

The practice of square sports entails loud music being played on speakers while several people dance or exercise under the direction of a leader. Participants see the square as a place where they can work out freely. However, non-participants think that square sports disturb public order, obstruct space, and create noise pollution [45,46,47,48].


*A square in a city is a place where people can refresh, but loud square-dancing songs are now played at the corners. So, the square, which should be quiet and relaxing, is no longer comfortable. In addition, in the community, people need to rest, but when they return home and lie down in bed, they are disturbed by loud music. As such, conflicts between residents and square dancers grew to the extent that they even fought.*



*We play the music loudly when we exercise in the square because those who practice at the back may not be able to hear it with everyone around. As a result, onlookers were greatly affected, and there were times when we quarreled with them because of the noise.*



*People exercise from 6 pm to 9 pm, but even those who do not exercise during this time walk and rest. Therefore, the noise or simply occupying public spaces makes those who do not exercise increasingly discontent. In recent news, there was a case in which square-dancing women fought with young men because they had occupied a basketball court.*


The above statements indicate that conflicts with non-participants intensify proportionally with the increase in participants. It is not an exaggeration to say that the main cause of the conflicts is the public character of the square. Although it is not owned by anyone and can be easily accessed, certain groups occupy the square, unexpectedly causing various disturbances for others, such as noise and traffic. This is the beginning of the conflict.

Non-participants express opposition in various ways, such as counterattacking high-pitched noise, scattering feces, shooting water bombs, and indiscriminately throwing iron beads [39]. Eventually, these conflicts escalate into physical fights, which cause significant property damage [48]. Not all square sports activities cause damage or inconvenience to the residents. Some people use government-owned squares far from residential areas and thus do not cause any nuisance. However, such alternative locations cannot accommodate many people, and many squares are located close to residential areas hindering dwellers from resting [5]. Recently, a few cities have adopted new regulations and alternative ways to reduce noise pollution. For example, dancing grannies are asked to use wireless headphones in the squares [46,48]. As such, appropriate measures are necessary to resolve the dissatisfaction of non-participants along with the enhancement of the participants’ social literacy.

### 3.2. Social Mechanism for Maintaining the Chinese Square Sports

Mechanism refers to the working principle, structure of things, or system within which the whole social structure operates. This study identified the factors necessary to understand the social mechanism by which square sports can be maintained and developed.

#### 3.2.1. People-Led Voluntary Participation

As the Chinese government allows dances and the expression of ethnic minority cultures as part of the “Healthy China” and “National Health” policies enacted in the 1990s, people voluntarily and enthusiastically participate in square sports to date [49].


*Our team has training principles, including voluntary participation, enjoying each day, trusting and helping each other, and acting amicably. As team members have these principles in mind, they help each other, and the atmosphere is good. Therefore, I would like to continue participating.*



*Square dancing is a voluntary exercise. All related expenses are borne by team members. All participants share the transportation and coaching expenses required for each square-dancing match. The costs are not high, and they are not burdensome.*


All study participants voluntarily participated in square sports and did not respond to governmental policies. They emphasized that “autonomy and spontaneity” play a critical role in activating square sports. This form of Chinese culture originated in group gymnastics, which began in the 1930s. In the chaotic social context of the time, when internal and external divisions occurred repeatedly, the state encouraged group gymnastics to strengthen its internal unity. After that, the People’s Republic of China, which advocated socialism, was established under Mao Zedong’s leadership in 1949. Group gymnastics became a crucial daily routine in people’s lives. At that time, it was a state-led activity to consolidate the foundation for national productivity, rather than to promote individuals’ health. After that, while implementing Deng Xiaoping’s reform and opening policy in 1978, the Chinese government emphasized that as it could not provide care for older adults, they should prepare for this stage of their lives by themselves through physical training. In other words, by encouraging group gymnastics to benefit individuals rather than the state, the Chinese government asserted that it was not a state-led activity.

According to the economic development plan of the Chinese government, collective physical activities began in squares in tandem with urban revitalization. The local governments spent significant amounts of money on building cultural squares that complemented each city’s characteristics to carry out Deng Xiaoping’s so-called “hold with both hands but hold tight” policy, which emphasized economic construction in conjunction with spiritual civilization and development [1]. Since the mid-1990s, squares have become a hot topic in China’s urban construction and an important means of embodying a city’s shape, displaying its history and culture, and promoting urban development. Because the country has a large population and stadium facilities are relatively scarce, people who want to participate in sports naturally flow toward squares, whose number is much greater.

As this activity became increasingly popular, the government enacted the “People’s Republic of China Sports Law” in 1995 to regulate it [50]. In 1998, it classified collective physical training groups as official groups, which it managed, and the rest as unofficial ones, to encourage the official registration of all such initiatives. However, the fact that 47.5% of the groups continue to be self-governing organizations shows that collective physical training cannot be fully subsumed into government regulation [14]. Eventually, it can be confirmed that collective physical activities led by the state were converted into square sports. It has been established as a people-led activity that is no longer subject to governmental management.

#### 3.2.2. Pride in Square Sports

Square sports create a new culture genre through their openness and inclusivity while enriching various living conditions, such as culture, tradition, politics, law, and social psychology through citizens’ participation [51]. Those who participate in square sports take great pride in making this a part of their lives and seeing it as a cultural heritage created over a long history.


*Square dancing is a product of a new cultural era that implies the obligation to provide opportunities for all people to become acquainted with each other, to help one another, and to be integrated into social and cultural life, as well as build a culture for society. We continued to have this culture.*



*Square dancing, which originated in folklore, is a combination of people’s lifestyles and production enhancement interventions. It is a cultural heritage that has been passed down through history in the form of a dance that people can enjoy on their own. The performance first occurred in the fields and moved into today’s parks and squares. It can be seen as a mixture of various ethnic dances. Although I am a teacher, I will do my best to continue developing this culture that bears 5000 years of the history of the Chinese people.*



*Our people are a nation that likes to dance together. When I go to the square to watch the dances of other races and also show the dances of our race, I feel that I am united with them. Only in this way will other people know that our people exist and that the culture of our people spread.*


The study participants perceived squares as places where various ethnic cultures could be shared in the form of art, beyond physical training. Additionally, as previously mentioned, square dancing has been created by the people. Since the founding of the People’s Republic of China, it has been continuously developing as it moved from rural areas to cities, becoming a distinctive folk art form [52]. In addition to square dancing, dance sports, aerobics, and jogging are also practiced in some places.

Through conversations with the study participants and the general public visiting the square, this study confirmed that they were highly proud of square sports, even though they were not participating. They were also confident that they represented an essential resource for Chinese culture.

From an anthropological perspective, culture has the property of being learned, stylized, and transmitted as a holistic lifestyle that embodies values and knowledge common to an era or group [53]. In this sense, square sports culture is an important carrier for constructing an urban spiritual civilization and has strong cohesiveness, thereby enriching the spiritual life of the public [36]. It is developing into a culture unique to China, as people learn and pass on the community’s values, beliefs, and cultural exchanges centered on physical training.

Thus, we conclude that Chinese square sports are not limited to the realm of physical activities but function as a means of forming and exchanging national cultures and solidifying national pride. Those things can be considered the mechanism of their continuous development.

#### 3.2.3. Reproduction of Economic Capital Using Human Resources

Leisure spaces can be created by individuals’ needs and actions but can also be shaped by capitalist development or government plans [54]. However, the spaces are socially structured whether leisure spaces are formed according to the government’s official plan and decision-making or by a group of individuals gathered according to their needs [22]. This study found that the squares were created as new leisure spaces, which enabled the participants to build a new capital.


*Sponsors were originally found among companies and brought by the leader. At the time, it was difficult to find a company. Since the state supports popular sports a lot, companies visit us first for promotion. They provide t-shirts and clothes in all four seasons.*



*First, sponsors were attracted in passive ways, whereby the leader, relatives, or friends looked for companies that wanted to advertise their products. However, sponsors voluntarily cooperate because the state advocates national health. Our team is now provided with clothes from Eastain stamped with the company’s name and logo.*



*Daming Glass sponsors clothes, which are printed with company advertisements. We received support from the company in the form of seasonal clothing. Every day, when team members exercise, they wear uniforms. The leader informs them about the clothes to wear each time.*


During the observation of the square sports practice, a team secured at least 50 participants wearing the same clothes sponsored by a company. However, teams with fewer than 30 members practicing folk dance, dance sports, or aerobics did not receive company sponsorship and enjoyed their activities wearing their own clothes. From Bourdieu’s perspective [55], agents produce a specific culture in the field for social activities. They have different amounts of resources and capital, which influence their position in the field. Similarly, this study identified that a company-sponsored team has more capital and takes up a bigger space than others. The sponsored team also takes an opportunity to gather people and consistently activate their space.

Thus, team size is an important factor for the eligibility of support.

As Figure 3 shows, companies sponsor sports players and teams to raise brand awareness and expand product sales. Market share is a strategy that has already become a standard procedure in sports marketing. However, Chinese square sports use a somewhat different strategy considering each city’s economic system. As society develops, the competition in each industry becomes increasingly fiercer. The geographical advantage of urban squares, exposed to many people, is emerging as an advertising resource. Each company further develops square sports activities through partnerships with local participants beyond providing support for various sports activities and events [51].

This study did not find conclusive evidence of the effects of corporate advertising among square sports participants. However, it led to the development and prosperity of some sports industries. In addition, it is noteworthy that this should continue because square sports have developed cultural productivity and people’s cultural livelihood as well as strengthened cultural soft power [36].

Eventually, Chinese square sports have gone beyond marketing the space termed square and established a new marketing strategy by combining human resources and corporate economic capital. Through this process, square sports have become the foundation for constructing a virtuous cyclical structure in which the Chinese sports industry thrives.

### 3.3. Limitations and Future Research Directions

The first limitation of this study is the difficulty in documenting the characteristics of approximately 6000 squares in China and the participants’ experiences. Therefore, the results cannot be generalized to all Chinese square sports. Second, this study focused on events that attracted many participants, such as square dance, walking, aerobics, and dance sports. The results might have been different if other events were examined.

We suggested that these shortcoming needs to be addressed in future studies. Future studies will consider other research approaches such as quantitative methods, mixed-research methods, photovoice, and multiple case studies to understand better a socio-cultural phenomenon and nature that occurred in sports squares. Furthermore, future studies will be needed to consider technological innovations to be utilized in square sports. Technology innovations allow people to enjoy and participate in various sports [56]. Sustainable technology innovation will also reduce global emissions and positively impact the environment. For example, creating App and virtual reality regarding square sports would offer solutions for conflicts that occupy space in sports squares. Especially using App will be a vital function in communicating with people participating in square sports and interacting continually among people. It will relieve conflict and motivate many people to participate in square sports. In light of the increasing number of smart cities worldwide, future researchers need to consider creating and developing technological innovation regarding square sports.

## 4. Conclusions

Spaces are mutually constructive zones that form and reproduce various social meanings, experiences, and identities beyond beings interacting with society [41]. Squares as spaces are part of the virtual environments in human lives. In particular, squares in downtowns exist everywhere in the world. They are meaningful spaces closely related to human life and the retainment of the history and characteristics of each country.

Chinese squares used to be places where government-led policies were practiced in the past, but they have now been transformed into centers for the public’s voluntary participation in various sports activities centered on square dancing. The latter has a long history, and participants enjoy the experience of overcoming multiple problems associated with urbanization through mutual exchanges and maintaining their own physical and mental health. Nevertheless, conflict with those who do not engage in square sports is emerging as a serious social problem. However, even non-participants consider square sports activities an important part of China’s culture and evoke a strong sense of pride. This means that they recognize that square sports function as an activity that enables the sharing of the cultures of ethnic minorities. In addition, as the number of companies supporting Chinese square sports increases, squares are no longer simply common leisure spaces for exercising and rest. They have become a foundation for the further development of this phenomenon as a function of the companies’ promotional and marketing initiatives. Therefore, the growth of Chinese square sports is expected to continue with these systems.

The triple bottom line by Elkington [57] is utilized as a framework for measuring corporate performance, corporate social responsibility, and sustainable development. The triple bottom line approach suggests considering economic, environmental, and social impact and measuring the three dimensions for sustainable social well-being, economic development, and environmental aspects [58,59,60]. In terms of economic sustainability, those participating in China’s Square Sports serve as a ‘living billboard’. This leads to positive aspects in the sense of exposure to local companies, diversification of corporate promotion, and enhancement of corporate brands. However, conflicts between teams participating in square sports are likely to damage corporate image and to regard square sports as companies’ promotional material. It would be possible to harm businesses and enterprises. As for environmental sustainability, Chinese square sports are a good example of the possibility of the coexistence of human beings and the urban environment. The square sports are not a spatial environment but a dynamic space with coexisting public and sports culture as well as historical value. However, social measures will be needed to resolve negative aspects including conflicts among people, noise pollution, and garbage occurring in square sports. Lastly, as for social sustainability, Chinese square sports became the pride of the local people due to their voluntary participation. People experience health recovery and maintenance and increase their quality of life through participating in square sports. Square sports contribute to social integration by sharing multiple cultures among local communities.

## Figures and Tables

**Figure 1 ijerph-19-15693-f001:**
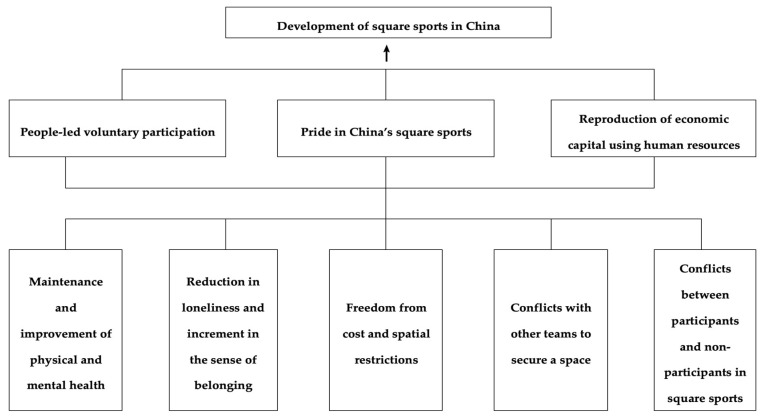
Presentation of Results.

**Figure 2 ijerph-19-15693-f002:**
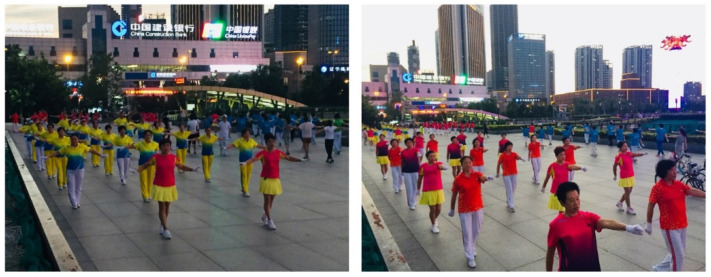
Image illustrating square sports through allocated places.

**Figure 3 ijerph-19-15693-f003:**
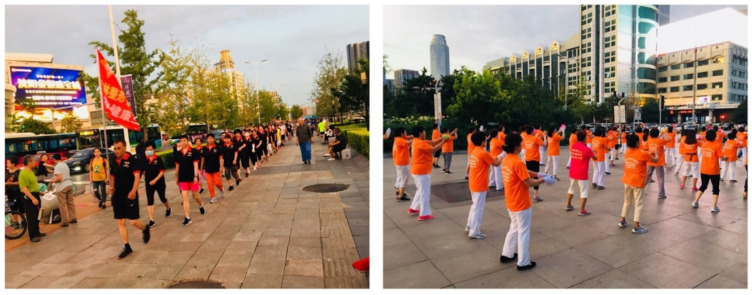
Image illustrating people who walk or dance with clothes sponsored by companies.

**Table 1 ijerph-19-15693-t001:** Study participants’ characteristics.

	Sex	Age (Year)	Occupation	Participation Experience (Year)	Event	Team Position
1	Female	57	Office worker	2	Square dancing	Participant
2	Female	52	Teacher	10	Square dancing	Participant
3	Female	66	Office worker	6	Dance sport	Participant
4	Female	53	Office worker	3	Square dancing	Participant
5	Male	48	Office worker	4	Walking	Participant
6	Male	58	Office worker	6	Walking	Leader
7	Female	54	Teacher	7	Square dancing	Leader
8	Female	55	Office worker	3	Dance sport	Leader
9	Female	55	Teacher	5	Square dancing	Leader
10	Male	65	Office worker	4	Square dancing	Leader

## Data Availability

No data were provided in this study.

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
