# Peer review of "What Is Happening in the Squares of China? Exploring the Experience of Participating in Square Sports and Sustainability Factors"

_ijerph, 2022, doi:10.3390/ijerph192315693_

Round 1
Reviewer 1 Report
Dear Authors,
I am pleased to review your paper. It is written well and looks reader-friendly. Moreover, it is quite informative and serves as a valuable dataset accompanied by analysis. My main concern is an insufficient literature review and a weak theoretical framework. Fortunately, it can be improved. I would suggest looking at squares from the perspective of sports innovation (see: Ekaterina Glebova & Michel Desbordes, 2021. "Technology innovations in sports: Typology, nature, courses and impact," Chapters, in Vanessa Ratten (ed.), Innovation and Entrepreneurship in Sport Management, chapter 5, pages 57-72, Edward Elgar Publishing. ) and smart city sports infrastructure (chapter 8 in https://www.goodfellowpublishers.com/academic-publishing.php?promoCode=&partnerID=/&content=story&storyID=459&fixedmetadataID=211 ).
Furthermore, the methodology should be explained and justified in a better way. Notably, the size of the sample (N=10) needs to be explained and proven as sufficient. Please provide a piece of detailed information about data validation. Furthermore, the phenomenological approach needs stronger justification and deeper explanation through the literature.
I hope these pieces of advice can be helpful to enhance your paper.
Reviewer 2 Report
This study aimed to explore the experience of participation in square sports in China and the social mechanisms by which they can be sustained. This is a topic of great interest. However, I have some comments, particularly of the quality of the English language and expressions. The manuscript needs thorough English editing. Additionally, social phenomena are complex and interactive. This study is qualitative in nature, requiring the authors to experience the square sports from an insider's perspective over a long time. However, did the authors use non-participatory, structured observations? If the participatory observation was used, please highlight it in the text.
Line 82-83: This sentence can be deleted.
Line 91-92: Suggest the author to check for errors. →“The purposeful sampling method was used to select study participants who could provide necessary information about a certain phenomenon.”
Line 92-94: Suggest to modify
Line 101-105: Suggest more rigorous language to describe
Line 119-122: A vague sentence.
Line 189: “Reduction” → “Reduce”
Line 234: “enjoy them” ? A vague subject.
Line 260: Who? This sentence doesn't have a subject.
Line 292: “Various”. And in the description of this study, the author should have distinguished the participants of the study from the other participants of the square event.
Line 393-394 & 438-439: “People's Republic of China Sports Law” Please cite references.
Line 550: “Limitations of the Study” →“Limitations”
Reviewer 3 Report
This paper aims to explore the experience of participating in square sports and sustaining the social mechanisms in China. The authors used qualitative phenomenological approach to interview 10 participants. The results showed that the participants’ physical and mental health were either maintained or improved as they engaged in square sports. The participants have also experienced reduced feelings of loneliness, an increase in their sense of belonging through conversing with other members of their teams. In addition, the participants were freed from the restrictions of cost and space in pursuing leisure activities. The social mechanisms of square sports were identified as people-led voluntary participation, pride in square sports, and the reproduction of economic capital using human resources. The following are my comments and suggestions to the authors.
1. Since this is qualitative research, the authors have to review more previous studies concerning the topics, including square sports, culture, quality of life, and sustainability. I didn’t find literature review for sustainability in the paper. Following literature review, the authors have to induce and deduce the main dimensions of the subject.
2. Environmental sustainability is related to ecology. Millennium Ecosystem Assessment (MA) uses a new conceptual framework for analyzing and understanding the effects of environmental changes on ecosystems and human wellbeing, putting the ecosystem services concept center stage. Ecosystem goods and services are classified into four categories: provisioning, regulating, supporting, and cultural services. I suggest that the authors may apply the concept of ecosystem services to construct the framework of the paper, which also contains social mechanisms.
3. In according to more literature review and reconstruct the framework, then out line can be drove for deep interview.
4. I suggest the authors should use up-to-date articles from 2020-2022.
Round 2
Reviewer 1 Report
Dear Authors,
Thank you for your efforts to address the recommendations and improve your paper. Yes, it still looks like there is room for further revisions as well. This paper is a promising piece of study, I would highly recommend focusing on revisions in order to deliver it to a reader. There are my suggestions:
1. (626-631) Could you please project future research perspectives for you and your colleagues? It seems logical if you turn your limitations into further research directions. Notably, seeing squares as an innovation and part of a smart city seems to be an interesting perspective, among others.
2. Sorry if I am wrong, but according to the authors' response it seems like the sports innovation concept is under-understood and neglected here. Just in case I attach the recommended piece of research for closer consideration and possible inclusion in the theoretical framework and future research directions. Please find it attached and kindly take your time to think square through an innovation perspective and part of a smart city. Definitely, it will increase the scientific soundness of the paper.
3. Thank you for extending the methods section and adding the validation info. However, it still needed to involve methodological kind literature on phenomenology and a clear statement of WHY this approach has been chosen by authors and HOW it helps to achieve the study's aims.

Author Response
Dear Reviewer 3
Thank you for providing valuable feedback and sharing professional material.
I have made an effort the revision based on your feedback. Please let me know if there are still rooms to be improved. I’d be happy to receive your feedback for improving our manuscript. The details are as follows:
Reviewer1’s suggestions:
- (626-631) Could you please project future research perspectives for you and your colleagues? It seems logical if you turn your limitations into further research directions. Notably, seeing squares as an innovation and part of a smart city seems to be an interesting perspective, among others.
- Just in case I attach the recommended piece of research for closer consideration and possible inclusion in the theoretical framework and future research directions. Please find it attached and kindly take your time to think square through an innovation perspective and part of a smart city. Definitely, it will increase the scientific soundness of the paper.
Response: Thank you again for sharing a file. It’s helpful to consider how a square is associated with ‘sports technology innovation.’ We added future research directions on p.14 and included future directions based on our study’s limitations and the article you offered. The below shows the writing for the revision.
(p. 14) We suggested that these shortcoming needs to be addressed in future studies. Future studies will consider other research approaches such as quantitative methods, mixed-research methods, photovoice, and multiple case studies to understand better a socio-cultural phenomenon and nature that occurred in sports squares. Furthermore, future studies will be needed to consider technological innovations to be utilized in square sports. Technology innovations make it possible for people to enjoy and participate in various sports [59]. Sustainable technology innovation will also contribute to reducing global emissions and positively impacting the environment. For example, creating App and virtual reality regarding square sports would offer solutions for conflicts that occupy space in sports squares. Especially, using App will be a vital function in communicating with people participating in square sports and interacting continually among people. It would also relieve conflict and motivate many people to participate in square sports. In light of the increasing number of smart cities worldwide, future researchers need to consider creating and developing technological innovation regarding square sports.
- It still needs to involve methodological kind literature on phenomenology and a clear statement of WHY this approach has been chosen by authors and HOW it helps to achieve the study's aims.
Response: I added a demonstration based on your guidance. Thank you!!
(p. 2) This study used qualitative phenomenological methods, which are the most suitable for achieving the purpose of this study. The phenomenological approach explores individuals’ experiential meaning and understands the lived world. The main point of the phenomenological method is to understand the commonalities of individuals’ experiences so that this focuses on what and how an individual experiences a specific phenomenon [21. 24]. Therefore, the phenomenological method is a critical approach to elucidate the meaning of lived experiences of a certain phenomenon for individuals with experiences. Given that this study aims to understand why people gather in squares, how they involve in square sports, and what square sports have been sustained, a phenomenological method is an appropriate approach to exploring a common phenomenon of what takes place in sports squares in China. In this regard, data were collected from study participants participating in square sports and detailed descriptions of the experiences shared with them were produced, regarding “what” they experienced and “how” they experienced it [20, 21, 24].
- We also added a description referring to Elkington’s framework of “Triple Bottom Line” in the conclusion. (p.15)
The triple bottom line by Elkington [60] is utilized as a framework for measuring corporate performance, corporate social responsibility, and sustainable development. The triple bottom line approach suggests considering economic, environmental, and social impact and measuring the three dimensions for sustainable social well-being, economic development, and environmental aspects [61, 62, 63]. In terms of economic sustainability, those participating in China's Square Sports serve as a ‘living billboard’. This leads to positive aspects in the sense of exposure to local companies, diversification of corporate promotion, and enhancement of corporate brands. However, conflicts between teams participating in square sports are likely to damage corporate image and to regard square sports as companies’ promotional material. It would be possible to harm businesses and enterprises. As for environmental sustainability, Chinese square sports are a good example of the coexistence possibility of human beings and the urban environment. The square sports are not a spatial environment but a dynamic space coexisting public and sports culture as well as historical value. However, social measures should be needed to resolve negative aspects including conflicts among people, noise pollution, and garbage occurring in square sports. Lastly, as for social sustainability, Chinese square sports became the pride of the local people due to their voluntary participation. People experience health recovery and maintenance and increase their quality of life through participating in square sports. Square sports contribute to social integration by sharing multiple cultures among local communities.
- We also arranged citations and references.
Thank you for your time for reviewing my manuscript.
I am looking forward to getting good news!
Sincerely.
Reviewer 2 Report
The authors responded well to the previous comments. The only minor issue is that the article's references should be rearranged. Correct, as suggested previously. I think it would be helpful to reinforce the findings by citing additional research already published.
Reviewer 3 Report
The authors also did not discuss the ecosystem services for sustainable development. Millennium Ecosystem Assessment (MA) uses a new conceptual framework for analyzing and understanding the effects of environmental changes on ecosystems and human well-being, putting the ecosystem services concept center stage. Ecosystem goods and services are classified into four categories: provisioning, regulating, supporting, and cultural services. I suggest that the authors may apply the concept of ecosystem services to construct the framework of the paper, which also contains social mechanisms.
Hope you are doing well.
